# Culturomics to Investigate the Endometrial Microbiome: Proof-of-Concept

**DOI:** 10.3390/ijms232012212

**Published:** 2022-10-13

**Authors:** Robin Vanstokstraeten, Shari Mackens, Ellen Callewaert, Susanne Blotwijk, Kristof Emmerechts, Florence Crombé, Oriane Soetens, Ingrid Wybo, Kristof Vandoorslaer, Laurence Mostert, Deborah De Geyter, Astrid Muyldermans, Christophe Blockeel, Denis Piérard, Thomas Demuyser

**Affiliations:** 1Department of Microbiology and Infection Control, Vrije Universiteit Brussel (VUB), Universitair Ziekenhuis Brussel (UZ Brussel), 1090 Brussels, Belgium; 2Brussels IVF, Vrije Universiteit Brussel (VUB), Universitair Ziekenhuis Brussel (UZ Brussel), 1090 Brussels, Belgium; 3Biostatistics and Medical Informatics Research Group (BISI), Vrije Universiteit Brussel (VUB), Laarbeeklaan 103, 1090 Brussels, Belgium; 4Center for Neurosciences, Faculty of Medicine and Pharmacy, Vrije Universiteit Brussel (VUB), Laarbeeklaan 103, 1090 Brussels, Belgium

**Keywords:** endometrial microbiome, culturomics, ART, embryo implantation, MALDI-TOF, 16S rRNA

## Abstract

The microbiome of the reproductive tract has been associated with (sub)fertility and it has been suggested that dysbiosis reduces success rates and pregnancy outcomes. The endometrial microbiome is of particular interest given the potential impact on the embryo implantation. To date, all endometrial microbiome studies have applied a metagenomics approach. A sequencing-based technique, however, has its limitations, more specifically in adequately exploring low-biomass settings, such as intra-uterine/endometrial samples. In this proof-of-concept study, we demonstrate the applicability of culturomics, a high-throughput culturing approach, to investigate the endometrial microbiome. Ten subfertile women undergoing diagnostic hysteroscopy and endometrial biopsy, as part of their routine work-up at Brussels IVF, were included after their informed consent. Biopsies were used to culture microbiota for up to 30 days in multiple aerobic and anaerobic conditions. Subsequent WASPLab^®^-assisted culturomics enabled a standardized methodology. Matrix-assisted laser desorption/ionization–time of flight mass spectrometry (MALDI-TOF MS) or 16S rRNA sequencing was applied to identify all of bacterial and fungal isolates. Eighty-three bacterial and two fungal species were identified. The detected species were in concordance with previously published metagenomics-based endometrial microbiota analyses as 77 (91%) of them belonged to previously described genera. Nevertheless, highlighting the added value of culturomics to identify most isolates at the species level, 53 (62.4%) of the identified species were described in the endometrial microbiota for the first time. This study shows the applicability and added value of WASPLab^®^-assisted culturomics to investigate the low biomass endometrial microbiome at a species level.

## 1. Introduction

In the last decades, increasing evidence has highlighted the presence of a unique microbiome not only in the lower, but also in the upper female reproductive system. Indeed, several studies have refuted the, hitherto assumed, sterile environment of the uterus and described distinct endometrial microbiota profiles [1]. The accumulating evidence suggests that the female reproductive tract is an open system with a continuum of microbiota gradually changing from the outer to the inner organs, with an increasing pH and a decreasing bacterial abundance from the vagina to the endometrium [2]. Yet, the number of well-powered clinical studies is still limited. As such, in contrast to the vaginal niche, a healthy or eubiosis-related endometrial microbiome is still not defined. The microbiota-immunity crosstalk seems of major importance in the upper female reproductive tract. Current theories suggest that a dysbiosis-related state triggers an inflammatory response in the endometrium that affects the embryonic implantation [3]. As such, a close relationship is suspected between the endometrial microbiome and conception, (sub)fertility, and pregnancy outcome [4,5,6].

A recent review article summarizes the findings of the currently available data with regard to female reproductive tract microbiota. The authors highlight the potential implication of these data in reproduction medicine and suggest further research and protocol standardization [7]. To date, all studies investigating the endometrial microbiome, apply 16S rRNA or metagenomics-based methodologies. Both approaches include whole genome sequencing (WGS) to analyze the genetic material present in the microbiome [8]. WGS enables a culture-free analysis of the microbiome, within a relatively short time and with limited effort. Despite the many advantages of this technique, there are also disadvantages to take into consideration.

Caution is specifically needed when sequence-based techniques are applied to study low biomass biological samples. DNA/RNA contamination in laboratory reagents and extraction kits can significantly bias the results of microbiota studies. This contamination is a concern for both rRNA gene sequencing requiring polymerase chain reaction (PCR) amplification, but also for shotgun metagenomics that do not require PCR. Many recent sequence-based studies describing microbial communities of low-biomass environments do not report the DNA/RNA quantification on the initial samples and do not include negative controls [9]. Another important drawback is the lack of standardization in the extraction and sequencing protocols [10]. Finally, a sequencing analysis cannot differentiate between living organisms and genetic fragments [11,12].

Because of the above-mentioned limitations of a sequencing-based microbiome analysis and the recent appreciation of the endometrial microbiome in female health and fertility, the application of alternative methodologies could be of relevance for upcoming studies and clinical practice. High-throughput culturing, or culturomics, combines a plethora of different enrichment broths, agar plates, and (an)aerobic incubation conditions to cultivate virtually all viable microbiota [13]. The cultured species are identified by matrix-assisted laser desorption/ionization time-of-flight mass spectrometry (MALDI-TOF MS) or 16S rRNA sequencing. Although culturing and MALDI-TOF MS are routinely used in a clinical laboratory setting and predate sequencing techniques, the high workload and poor traceability limit their application in microbiome research. Yet, culturomics-obtained microbiome data can be complementary to the metagenomics approach [14]. With the use of several culturing media and the inclusion of adequate negative controls, the depth and DNA-extraction bias of a metagenomics-based microbiome analysis are bypassed in culturomics. A major advantage of the culturing approach is that viable bacteria and fungi are identified on a species level. In addition, microorganisms present at a very low biomass level can be detected, allowing for the detection of minority populations that potentially have a substantial effect on the ecology of microbiota [12]. Even though, culturomics provides additional information, this information is only available in qualitative terms, where sequencing approaches also give a representation of the quantitative composition of microbiota.

Previous studies highlight certain challenges in a successful culturomics set-up. Fastidious anaerobic species, such as *Fusobacterium nucleatum* and *Peptostreptococcus anaerobius*, often grow very slowly, require special nutrients, are rapidly grown over by other species, and die in the presence of oxygen [15]. As a result, most of these species are not suitable for standard cultivation on agar plates. A primary incubation in an anaerobic liquid enrichment medium provides a suitable environment for the growth of these species. The subsequent agar inoculation from these broths enables the detection of these slow-growing anaerobes [14]. Compared with other ecosystems, culturomics has mostly been applied to the human gut microbiota. However, this technique also showed its utility on vaginal microbiota. Since the application of culturomics at least 15 new species from the vaginal tract haven been reported [13,15].

In the current proof-of-concept study, we applied a culturomics-based analysis of endometrial samples. The enrichment and incubation conditions were based on previously described protocols [14]. To lower the workload and standardize the culturomics approach, we applied a semi-automated inoculation and incubation system: WASPLab^®^ (Copan Diagnostics, Brescia, Italy). As such, we are the first to describe a culturomics-based endometrial microbiome profile.

## 2. Results

Ten endometrial biopsies were included in the current study. The characteristics of the women are depicted in Table 1. A total of 2933 colonies were identified using the MALDI Biotyper^®^ system. Eighty-five different microorganisms are listed in Figure 1 and Figure 2 and Table 2, including two fungi (*Candida glabrata* and *Candida parapsilosis)*. Twenty-six of these bacterial isolates (31%) were Gram-negative and 33 (39%) were obligate anaerobes. These 85 species belonged to 40 different genera and 28 different families. Four species *(Lactobacillus coleohominis, Prevotella colorans, Porphyromonas bennois* and *Paenibacillus xylanilyticus*) were not identifiable with MALDI-TOF MS and were identified using 16S rRNA gene sequencing. One bacterial isolate could only be identified on the genus level (*Demequina* species). *Corynebacterium, Lactobacillus, Prevotella* and *Staphylococcus* are the best-represented genera among the ten samples in terms of the diversity in the species, as we identified seven species within each of them.

*Stenotrophomonas maltophilia* was found in one of our negative control samples. Therefore, the observation of this species was excluded from the analysis. The mass spectrometry used in this study cannot differentiate between *Peptoniphilus harei* and *Peptoniphilus indolicus.* Therefore, in subsequent analyses, they have been treated as one species as this was considered to cause less of an error than leaving them out completely.

In an average endometrial biopsy, 18.6 different species were cultured (8.6 with direct inoculation and 12.9 with pre-incubation). The minimum amount of species found in a sample was seven, the maximum was 41. A very similar number of species were found without the pre-incubation, with the aerobic pre-incubation or with the anaerobic pre-incubation. This holds true both for the absolute number of species, as well as the unique number of species not found in either of the other two incubation options, illustrated in Figure 1, Figure 2, and Table 3. The different incubation-techniques all contribute substantially to the identification of the most clinically relevant species, such as the various *Lactobacillus* and bacterial vaginosis and aerobic vaginitis-related species (Appendix A). Yeasts, however, were exclusively found after pre-incubation.

As can be expected, there is quite some overlap in the species found within each sample, for each technique. Nevertheless, they each contribute enough additional information to justify performing each, in addition to the other two techniques (Figure 3 and Figure 4). Removing any of the three incubation options in this study would have led to an average loss of 20% to 25% of the species found per sample. Surprisingly, the number of obligate anaerobic species found in the anaerobic pre-incubation was slightly lower than in the aerobic pre-incubation (Figure 5). For example, *Anaerococcus murdochii, Peptoniphilus duerdenii, Peptostreptococcus anaerobius* and *Prevotella buccalis* are obligate anaerobes and were only identified using aerobic pre-incubation.

It appears that a non-negligible part of the endometrial microbiome consists of slow growing species, as almost one sixth of the species found only showed up after 30 days of pre-incubation (Figure 5). *Actinomyces radingae, Bacteroides coagulans, Demequina* species, *Dermacoccus nishinomiyaensis, Paenibacillus xylanilyticus, Porphyromonas bennonis, Prevotella colorans* and *Trueperella bernardiae* were only identified after 30 days of pre-incubation.

Aside from the position of sample one, clustering was fairly consistent across single, average, and complete linkages. There appears to be some correlation between patients’ age and the clusters they are in, although this is most likely a result of random chance (Figure 6).

## 3. Discussion

Culturomics has proven to be a very valuable tool in exploring human microbiota and has vastly enhanced our understanding of it [13]. This is the first time the approach has been applied to endometrial biopsies, despite the very high need in expanding the knowledge about it. In this study, we conducted culturomics with the help of total lab automation (TLA) on 10 endometrial biopsies. The data show that, in strong contrast to what has been reported in other studies [17], many microorganisms out of the endometrium can be cultured. Furthermore, we demonstrated that the TLA systems, as for example the WASPLab^®^ used in our study, could be valuable tools in tackling the extremely high workload of culturomics and provide standardization and a perfect traceability.

Based on the review article of *Punzón* et al. [7], we can report a good concordance with the previous described metagenomics-based endometrial microbiota in subfertile and infertile women. Seventy-seven (91%) of the identified species belonged to previously described genera, present in the endometrial microbiome. We described 53 species and seven genera for the first time in the endometrial microbiome. However, we did not find species belonging to *Acinetobacter* or *Pseudomonas*, two of the most frequently described genera in the endometrial microbiome, based on sequencing. This may be explained by the fact that the species within the genera *Acinetobacter* and *Pseudomonas* are known to be contaminants of DNA-extraction kits, DNA-free water, PCR primers and other laboratory reagents. This contamination is a real challenge to study low-biomass microbiota such as, the endometrial microbiome, which provide very little template DNA to compete with that in the reagents for sequencing. The importance of this issue when analyzing low biomass samples, despite the multiple reports of reagent contamination, remains underappreciated in the microbiota research community [9]. A culture-based approach such as the one we used, could overcome these biases.

Interestingly, we found more obligate anaerobic species using the aerobic pre-incubation bottle than with the anaerobic pre-incubation bottle. We cannot give an unequivocal explanation for this, but it was notable that the *Lactobacillus* species often grew much more abundantly in the anaerobic bottle than in the aerobic bottle. This lowers the pH, which inhibits the growth of some anaerobic bacteria [18]. Finally, the aerobic atmosphere in the aerobic bottle is only temporary, since all of the oxygen present in the bottle is used after a few days of incubation. Plate streaking is a major part of the culturomics workload. The inoculation procedures critically influence the specimen identification, mainly by the ability to generate single colonies. When performing plate streaking manually, reproducibility is poor. When using WASPLab^®^ for plate streaking, a perfect reproducibility is almost guaranteed. The automated plate inoculation will also lead to a higher frequency of single-colony recovery, resulting in an increased detection of species. On top of that, inoculation with WASP^®^ is time-saving: Quiblier et al. showed 25 min saved “hands-on time” for every 100 plates streaked [19]. Following the inoculation, there is a swift and automatic transfer into the incubators with a more stable temperature control than classic incubators. A perfect traceability is guaranteed due the unique label each streaked plate is given by the system.

Culture-based studies on endometrial samples are rare. Only very few studies describe the use of culture techniques on endometrial samples, almost all of them looking for very specific and possible pathogenic species, such as *Escherichia coli* and *Gardnerella vaginalis*. To our knowledge, only Smolnikova et al. tried to culture endometrial samples with extended nutrient non-selective media [16]. In their study, they enrolled 80 patients with fertility problems. Following a routine embryo transfer, the most distal five mm portion of the embryo transfer catheter was used for the culture with extended nutrient media. However, these nutrient media were not defined in their article. Disseminated over these 80 samples, they managed to identify 33 different bacterial species. In contrast with our culturomics approach, they established a low bacterial diversity: 28% of the endometrial samples showed a monoculture and 12% of the samples showed no bacterial growth at al. Of the 33 species they described, we found 21 (64%) back in our study. They described that the *Lactobacillus* species were the most frequently isolated, with a frequency of 80%. We confirmed this observation as we isolated the *Lactobacillus* species in all 10 samples, with *Lactobacillus jensenii* as most commonly isolated species. Studies based on metagenomics also confirm the significant part of *Lactobacilli* in the endometrial microbiome, suggesting *Lactobacillus* as the most represented genus in a endometrial microbiome [7]. However, there is still no consensus. Additional findings of our study highlight the importance of aerobic and anaerobic incubation, with and without pre-incubation, as dysbiosis-related strains seem to dominate without pre-incubation enrichment (Appendix A). As such, the inclusion of different types of sample incubation conditions are very important to achieve a non-biased culture-based approach on the microbiota composition of endometrial samples.

The first drawback of this study is the relatively small amount of used culture conditions that were selected, based on previous studies performed on stool samples [14]. Despite the high bacterial diversity resulting from our bacterial culture media, a more extensive bacterial cultivation would probably expand the amount of identified species. Although being very challenging, it could be interesting for subsequent studies to take into account the physicochemical properties of the endometrium when selecting additional culture conditions. For example, one could experiment with pH, temperature and hormones: three important and dynamic factors in the endometrium [20]. These factors are variable across the follicular, ovulatory and luteal phases, resulting in dynamic uterine microbiota during the menstrual cycle [21]. Although the selection of additional culture media has to be considered very carefully and is not directly proportional to the discovery of additional species, such as was highlighted by Diakite et al. One of the most important factors is probably the use of rumen fluid in the blood culture bottles, which showed to have a great impact in the isolation of strict anaerobic bacteria [14]. A second and insurmountable drawback of this study is the possible sample contamination with microorganisms originating from the lower reproductive tract. The impact of various contaminating interventions makes it hard to only detect the low-abundance microorganisms originating from the sampling site [22]. Indeed, due to the extremely high biomass in the lower reproductive tract, compared to the uterine microbiome and the transcervical approach that cannot be avoided, the misrepresentation of the species distribution cannot be excluded. We have to keep in mind that surgical manipulations and instruments disrupting the cervical barrier, such as, for example a hysteroscopy, could carry over traces from the lower reproductive tract to the uterus. However, the exact impact of these manipulations is unclear and needs to be further investigated. For example, by comparing cultures of endometrial biopsies before and after a hysteroscopy. The influence of the blood microbiome should also be taken into account in subsequent studies, as it seems likely that some types of microorganisms could move to the endometrium via the blood [23]. Then, we have to keep in mind that only a relatively small number of patients was included. Because 38 of the discovered species were found in only one of the 10 samples, the number of species would probably increase strongly when expanding the study-population. As such, the study-population is too small to consider results from clustering in any way representative of the general population. While no conclusions should be drawn from any specific clusters in this data set, it is clear that the samples are quite informative, as there is sufficient variance and range in the distances. This is partly due to the fact that the distances can be calculated down to the species level. Therefore, it clearly illustrates the potential of culturomics for differentiating between microbiota relative to certain clinical parameters.

Finally, no additional amplicon/metagenome analysis was performed on our own samples. Due to the very small volume of the biopsy, we chose to use the whole sample for culture. We tried to compensate for this missing part by comparing our results with the literature (similar tissues, different techniques).

## 4. Material and Methods

### 4.1. Setting and Study Design

Ten subfertile women, undergoing diagnostic hysteroscopy followed by an endometrial biopsy, as part of the routine work-up at Brussels IVF, Universitair Ziekenhuis Brussel, were included in this study. Following the hysteroscopy, an endometrial biopsy was taken with a sterile inner-outer catheter minimalizing vaginal and cervical microbial contamination. All biopsies were taken during the follicular phase. The biopsy was collected and transported within minutes to the microbiology laboratory in an eSwab^®^ tube (Copan Diagnostics, Brescia, Italy) to minimalize the possible death of obligate anaerobe microorganisms [24].

### 4.2. Culturomics

The culturomics setup was based on a previous publication [14]. Sample processing is schematically summarized in Figure 7.

#### 4.2.1. Direct Inoculation

In a laminar flow cabinet, 1.0 mL of sterile 0.9% NaCl was added to the sample in order to increase the volume before homogenizing the biopsy through a sterile pestle and mortar. Once homogenized, a first part of the sample is manually inoculated on multiple aerobic and anaerobic agar plates and broths. The following culture media were used for the direct inoculation: MacConkey agar, aerobic blood agar, anaerobic agar, selective anaerobic agar, Sabouraud agar, Schaedler agar, chocolate agar PolyViteX VCAT3, *Ureaplasma/Mycoplasma* agar, *Ureaplasma* broth and *Mycoplasma* broth (Table 4). For this, we inoculated a 30 µL sample per agar plate and two drops of the sample per broth. Following five days of incubation at 37 °C in conventional aerobe and anaerobe incubators, the colonies on the plates are identified using the MALDI-TOF MS. If the MALDI-TOF MS failed in identifying the strain, a 16S rRNA gene sequencing was performed. All of the above-mentioned actions took place in an aerobic atmosphere at room temperature. The broths were assessed visually: a green color in the *Ureaplasma* broth indicated the growth of the *Ureaplasma* species, a red color in the *Mycoplasma* broth indicated the growth of *Mycoplasma*.

#### 4.2.2. Pre-Incubation

The second part of the homogenized sample was used for the pre-incubation. Two different pre-incubation conditions were applied. For each sample, we enriched both an aerobic and an anaerobic blood culture bottle (BACT ALERT ^®^ FA and FN Plus, Marcy-l’Étoile, France) with 2.0 mL sterile rumen fluid, 2.0 mL sterile defibrinated sheep blood and 2.0 mL of a homemade supplement mix (see below). Both enriched blood culture bottles were inoculated with a 0.5 mL sample using sterile syringes and needles working in a laminar flow cabinet. The above-mentioned actions were carried out in an aerobic atmosphere at room temperature. The enriched blood culture bottles were incubated at 37 °C.

The inoculation on multiple aerobic and anaerobic agar plates, from the pre-incubation media, was performed at one, five, 10, and 30 days post-incubation. One mL of the pre-incubated medium was added to a Vacuette tube (Greiner, Alphen aan den Rijn, The Netherlands). These tubes and agar plates were provided to WASPLab^®^ (Copan Diagnostics, Brescia, Italy) for the automatic inoculation. This WASPLab^®^ system contains a specimen processor, culture incubator and work-up system [25]. The following agar plates were used after the pre-incubation: MacConkey agar, aerobic blood agar, anaerobic agar, selective anaerobic agar, Sabouraud agar, Schaedler agar and chocolate agar PolyViteX VCAT3 (Table 4).

The inoculation was performed with a 1.0 µL loop using the five-fold dilution-striking pattern. The automatic CO_2_ incubator and non-CO_2_ aerobic incubator of our WASPLab^®^ system were used after the inoculation. Due the fact that WASPLab^®^ does not offer an anaerobic incubator, these plates were placed manually in our anaerobic incubator after inoculating with the WASP^®^. Five days after the incubation, all of the different bacterial and fungal colonies were isolated for MALDI-TOF MS or 16S rRNA sequencing. Blank aerobic and anaerobic blood culture bottles, both supplemented with 2.0 mL sterile rumen fluid, 2.0 mL sterile defibrinated sheep blood and 2.0 mL of our homemade supplement mix, accompanied every sample throughout the whole process to ensure the sterility of the used substances. All samples and incubation media were traced through a barcoding system provided by WASPLab^®^. WASPLab^®^ settings and incubators are summarized in Table 5.

#### 4.2.3. Rumen Fluid

Rumen fluid was obtained post-mortem from a freshly slaughtered cow and sterilized by a cascade of filtration and centrifugation steps. No autoclave was used in order to preserve the thermolabile components. The sterility was checked at the end of the process. The sterilization process is summarized in Figure 8.

#### 4.2.4. Supplement Mix

The composition of the supplement mix is based on the YCFA and the R-medium, previously used by Didier Raoult et al. [14]. Sixteen different ingredients were used. Twelve ingredients are sterilized through an autoclave, four ingredients are thermolabile and were therefore sterilized through filtration. The sterility was checked at the end of the process. All components are summarized in Table 6.

#### 4.2.5. Identification of the Colonies

Identification of the colonies was performed using a MALDI-TOF MS Biotyper^®^ (Bruker Daltonics, Bremen, Germany) system. Based on the morphology, a pure colony was selected and spotted on a polished steel MALDI MSP 96 target (Bruker Daltonics, Bremen, Germany). The spot is overlaid with one µL formic acid. Once air-dried at room temperature, the spot is overlaid with one µL matrix solution of α-cyano-4-hydroxycinnamic acid (Bruker Daltonics, Bremen, Germany): 10 mg/mL in standard solvent solution (50% acetonitrile, 47.5% water and 2.5% trifluoroacetic acid). The spot is dried at room temperature. The samples were analyzed using the MALDI Biotyper^®^ Sirius system (Bruker Daltonics, Bremen, Germany) equipped with the Smartbeam MBT version GFLIC-2 laser-positive mode. The spectra were collected using manufacturer software, FlexControl version 3.4 build 207.20. The following parameters (MBT-AutoX method) were used: 240 shots in-one spectra, attenuator offset = 54%, attenuator range = 20%, initial laser power 30%, maximum laser power 40% and frequency = 200 Hz.

The isolates with an identification score > 2.0 with a unique hit were considered as accurately identified using the MALDI MBT compass 4.1 build 100. The isolates with an identification score < 2.0 were preserved at −80 °C for 16S rRNA sequencing. Eurofins Genomics (Ebersberg, Germany) performed the 16S rRNA sequencing on the nine non-identifiable isolates. BioNumerics v.7.6.3 (Applied Maths, Sint-Matens-Latem, Belgium) was used for the assemblage of the sequenced data. The assembled data was analyzed using the Basic Local Alignment Search Tool (BLAST) available on the U.S. National Library of Medicine (https://blast.ncbi.nlm.nih.gov/ accessed on 1 August 2022). To describe an isolate on the species level, ≥99.0% identity with a known sequence and ≥0.8% difference with the second best match is needed [26].

#### 4.2.6. Statistics

The parameter estimates and confidence intervals were obtained from the Poisson GLMM repeated measures analyses using the R package ‘Ime4’ [27]. The unweighted UniFrac distance was used as a beta diversity metric for agglomerative clustering.

## 5. Conclusions

The need in understanding the endometrial microbiome and its influence on a successful pregnancy is very high. In this study, we showed that the use of culturomics could give novel insights into the endometrial microbiome and that WASPLab^®^ is a valuable tool in processing the extremely high workload.

## Figures and Tables

**Figure 1 ijms-23-12212-f001:**
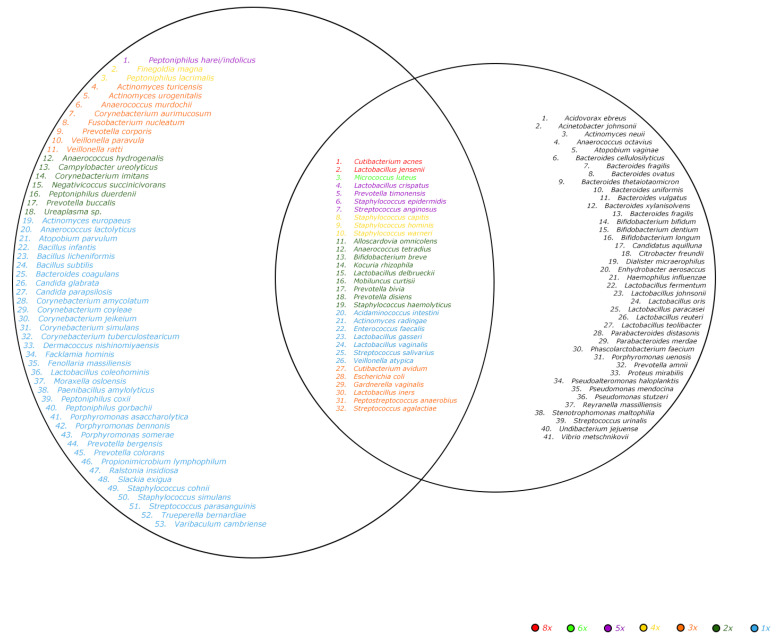
Venn-diagram of described species in the endometrium microbiome. Left circle: species found with our culturomics protocol. Right circle: species described in the literature [7,16]. Inner circle: species found in both our study and the literature [7,16]. The color describes the prevalence over the 10 samples ranging from present in one sample to present in eight samples. Literature overview includes both infertile and subfertile women.

**Figure 2 ijms-23-12212-f002:**
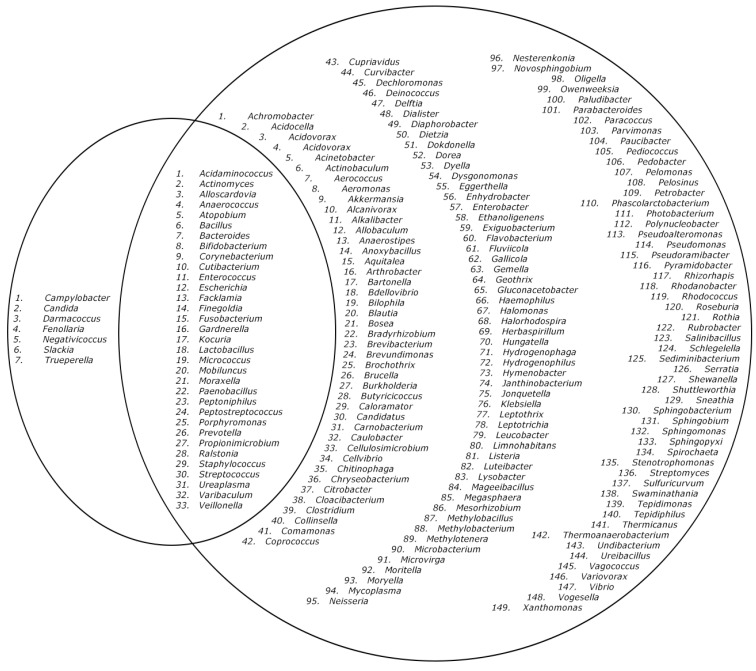
Venn-diagram of the described genera in the endometrium microbiome. Left circle: genera found with our culturomics protocol. Right circle: genera described in the literature [7,16]. Inner circle: genera found in both our study and the literature [7,16]. Literature overview includes both infertile and subfertile women.

**Figure 3 ijms-23-12212-f003:**
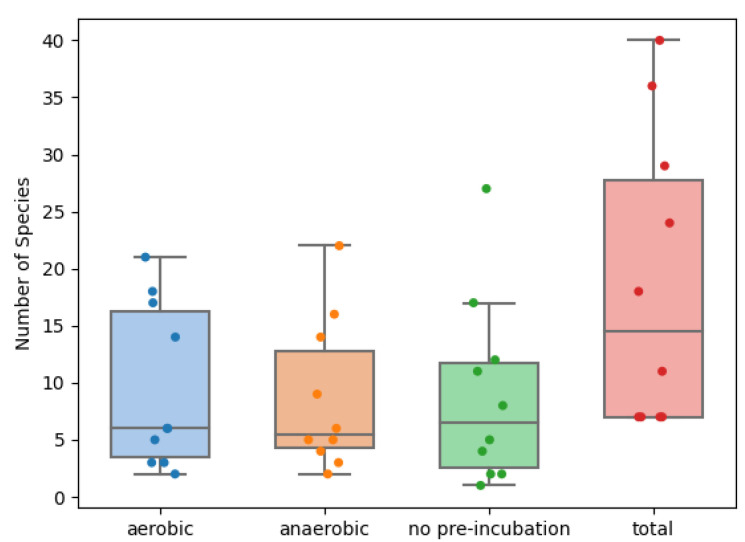
Scattered boxplot for the total number of species found per sample per incubation method (blue: aerobic pre-incubation, orange: anaerobic pre incubation, green: no pre-incubation and red: total).

**Figure 4 ijms-23-12212-f004:**
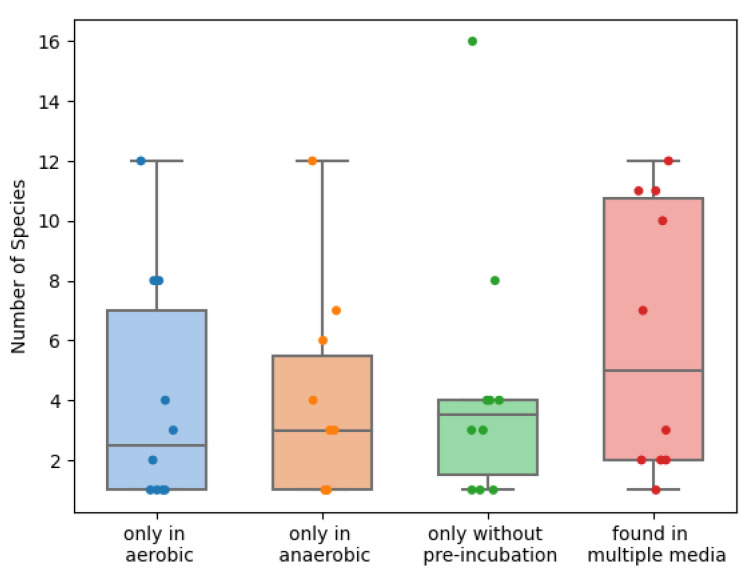
Scattered boxplot for the number of unique species found within each sample per incubation method (blue: aerobic pre-incubation, orange: anaerobic pre incubation, green: no pre-incubation and red: total).

**Figure 5 ijms-23-12212-f005:**
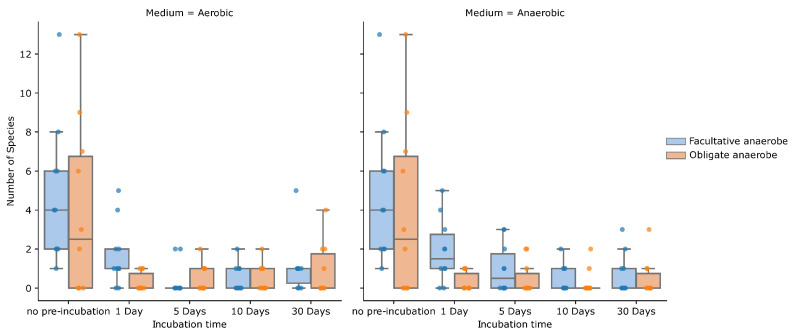
Scattered boxplot of the number of facultative (blue) and obligate (orange) anaerobic species at different incubation times.

**Figure 6 ijms-23-12212-f006:**
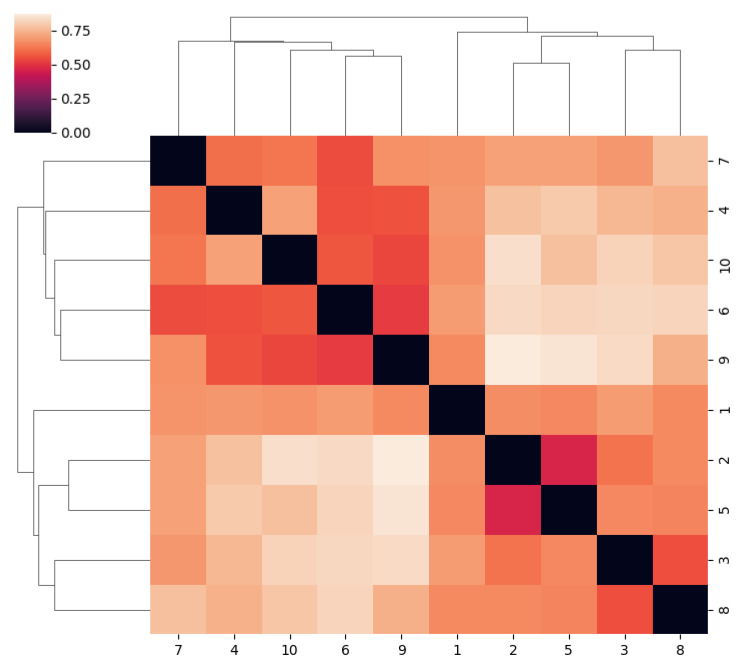
Heatmap of the unweighted UniFrac distances between the samples, along with a dendrogram of the agglomerative hierarchical clustering with a complete linkage method. The numbers from one to 10 on the X and Y axis represent the patient samples. The color in the figure represents phylogenetic relatedness, darker colors equals closer relatedness.

**Figure 7 ijms-23-12212-f007:**
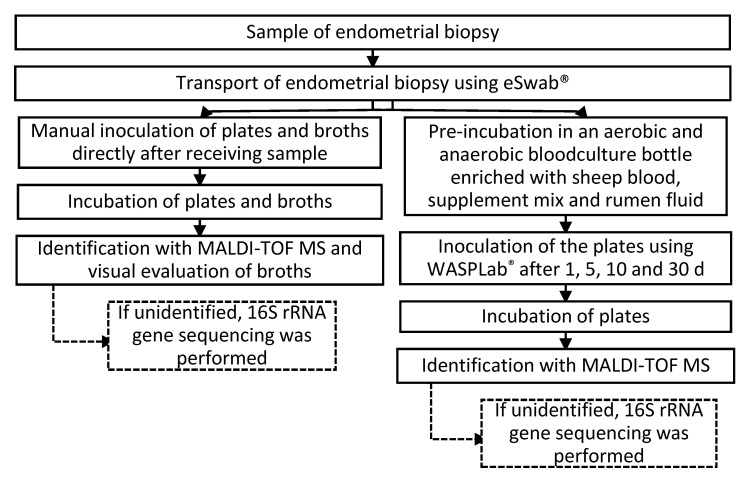
Summary of our culturomics workflow. The endometrial sample is collected and transported in an eSwab^®^ tube. Following the homogenization of the biopsy with a sterile pestle and mortar, the sample is partly manually inoculated on agar plates. Next, five days later, the colonies on the plates are identified using the MALDI-TOF MS. The other part of the sample is injected in two pre-incubation bottles and incubated for 30 days. Then, after 1, 5, 10 and 30 days, 1 µL of the bottle is inoculated on the agar plates with the WASPLab^®^. r Five days later, the colonies on the plates are identified using the MALDI-TOF MS. If the MALDI-TOF MS fails in identifying the strain, a 16S rRNA gene sequencing was performed.

**Figure 8 ijms-23-12212-f008:**
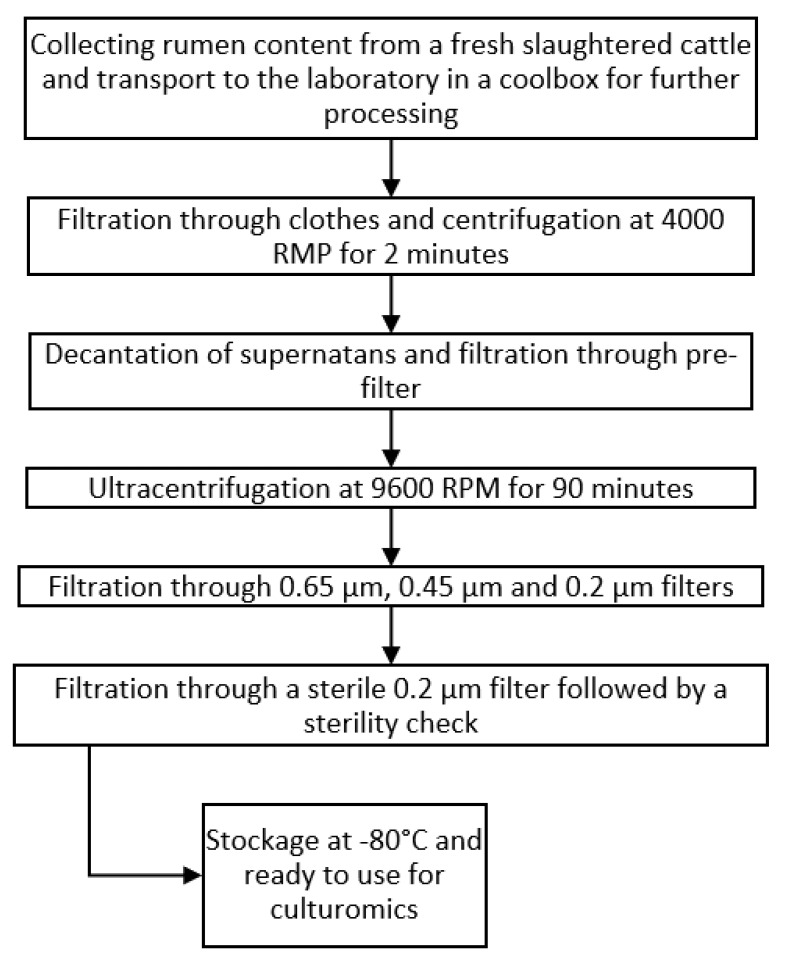
Workflow we used for sterilizing the rumen fluid. No autoclave was used in order to preserve the thermolabile components. Sterility was checked at the end of the process.

**Table 1 ijms-23-12212-t001:** Clinical data from the 10 patients enrolled in the study. Observations of the vagina, cervix, cavum and ostium were documented in the context of their fertility traject. From every patient, a biopsy was analyzed in the laboratory of anatomopathology for the presence of plasma cells indicating inflammation.

Patient	Age (Years)	Phenotype	Vagina	Cervix	Cavum	Left Ostium	Right Ostium	Anatomopathology
1	34	European	Normal	Normal	Polypoid	Normal	Normal	Normal histology
2	31	European	Normal	Normal	Polyp	Polyp	Polyp	Normal histology
3	26	North-African	Normal	Normal	Normal	Normal	Normal	Normal histology
4	46	European	Normal	Long cervix	Normal	Normal	Normal	Isolated plasma cells
5	25	European	Normal	Normal	Mullerian anomaly: septum	/	/	Isolated plasma cells
6	41	North-African	Normal	Normal	Normal	Normal	Normal	Normal histology
7	44	European	Normal	Normal	Polyp	Obstruction	Obstruction	Normal histology
8	30	North-African	Normal	Normal	Normal	Normal	Normal	Isolated plasma cells
9	29	European	Normal	Normal	Normal	Normal	/	Normal histology
10	30	North-African	Normal	Normal	Normal	Normal	Normal	Normal histology

**Table 2 ijms-23-12212-t002:** Summary of the 85 different species found in the 10 endometrial biopsies. Detailed information per patient and per species available in Appendix A.

Species	Phylum	Family	Genus
*Acidaminococcus intestini*	Firmicutes	Acidaminococcaceae	*Acidaminococcus*
*Actinomyces europaeus*	Actinobacteria	Actinomycetaceae	*Actinomyces*
*Actinomyces radingae*	Actinobacteria	Actinomycetaceae	*Actinomyces*
*Actinomyces turicensis*	Actinobacteria	Actinomycetaceae	*Actinomyces*
*Actinomyces urogenitalis*	Actinobacteria	Actinomycetaceae	*Actinomyces*
*Alloscardovia omnicolens*	Actinobacteria	Bifidobacteriaceae	*Alloscardovia*
*Anaerococcus hydrogenalis*	Firmicutes	Peptoniphilaceae	*Anaerococcus*
*Anaerococcus lactolyticus*	Firmicutes	Peptoniphilaceae	*Anaerococcus*
*Anaerococcus murdochii*	Firmicutes	Peptoniphilaceae	*Anaerococcus*
*Anaerococcus tetradius*	Firmicutes	Peptoniphilaceae	*Anaerococcus*
*Atopobium parvulum*	Actinobacteria	Atopobiaceae	*Atopobium*
*Bacillus infantis*	Firmicutes	Bacillaceae	*Bacillus*
*Bacillus licheniformis*	Firmicutes	Bacillaceae	*Bacillus*
*Bacillus subtilis*	Firmicutes	Bacillaceae	*Bacillus*
*Bacteroides coagulans*	Firmicutes	Peptoniphilaceae	*Bacteroides*
*Bifidobacterium breve*	Actinobacteria	Bifidobacteriaceae	*Bifidobacterium*
*Campylobacter ureolyticus*	Proteobacteria	Campylobacteraceae	*Campylobacter*
*Candida glabrata*	Ascomycota	Saccharomycetaceae	*Candida*
*Candida parapsilosis*	Ascomycota	Saccharomycetaceae	*Candida*
*Corynebacterium amycolatum*	Actinobacteria	Corynebacteriaceae	*Corynebacterium*
*Corynebacterium aurimucosum*	Actinobacteria	Corynebacteriaceae	*Corynebacterium*
*Corynebacterium coyleae*	Actinobacteria	Corynebacteriaceae	*Corynebacterium*
*Corynebacterium imitans*	Actinobacteria	Corynebacteriaceae	*Corynebacterium*
*Corynebacterium jeikeium*	Actinobacteria	Corynebacteriaceae	*Corynebacterium*
*Corynebacterium simulans*	Actinobacteria	Corynebacteriaceae	*Corynebacterium*
*Corynebacterium tuberculostearicum*	Actinobacteria	Corynebacteriaceae	*Corynebacterium*
*Cutibacterium acnes*	Actinobacteria	Propionibacteriaceae	*Cutibacterium*
*Cutibacterium avidum*	Actinobacteria	Propionibacteriaceae	*Cutibacterium*
*Dermacoccus nishinomiyaensis*	Actinobacteria	Dermacoccaceae	*Darmacoccus*
*Enterococcus faecalis*	Firmicutes	Enterococcaceae	*Enterococcus*
*Escherichia coli*	Proteobacteria	Enterobacteriaceae	*Escherichia*
*Facklamia hominis*	Firmicutes	Aerococcaceae	*Facklamia*
*Fenollaria massiliensis*	Firmicutes	Eubacteriales	*Fenollaria*
*Finegoldia magna*	Firmicutes	Peptoniphilaceae	*Finegoldia*
*Fusobacterium nucleatum*	Fusobacteria	Fusobacteriaceae	*Fusobacterium*
*Gardnerella vaginalis*	Actinobacteria	Bifidobacteriaceae	*Gardnerella*
*Kocuria rhizophila*	Actinobacteria	Micrococcaceae	*Kocuria*
*Lactobacillus coleohominis*	Firmicutes	Lactobacillaceae	*Lactobacillus*
*Lactobacillus crispatus*	Firmicutes	Lactobacillaceae	*Lactobacillus*
*Lactobacillus delbrueckii*	Firmicutes	Lactobacillaceae	*Lactobacillus*
*Lactobacillus gasseri*	Firmicutes	Lactobacillaceae	*Lactobacillus*
*Lactobacillus iners*	Firmicutes	Lactobacillaceae	*Lactobacillus*
*Lactobacillus jensenii*	Firmicutes	Lactobacillaceae	*Lactobacillus*
*Lactobacillus vaginalis*	Firmicutes	Lactobacillaceae	*Lactobacillus*
*Micrococcus luteus*	Actinobacteria	Micrococcaceae	*Micrococcus*
*Mobiluncus curtisii*	Actinobacteria	Actinomycetaceae	*Mobiluncus*
*Moraxella osloensis*	Proteobacteria	Moraxellaceae	*Moraxella*
*Negativicoccus succinicivorans*	Firmicutes	Veillonellaceae	*Negativicoccus*
*Paenibacillus amylolyticus*	Firmicutes	Paenibacillaceae	*Paenobacillus*
*Peptoniphilus coxii*	Firmicutes	Peptoniphilaceae	*Peptoniphilus*
*Peptoniphilus duerdenii*	Firmicutes	Peptoniphilaceae	*Peptoniphilus*
*Peptoniphilus gorbachii*	Firmicutes	Peptoniphilaceae	*Peptoniphilus*
*Peptoniphilus harei/indolicus*	Firmicutes	Peptoniphilaceae	*Peptoniphilus*
*Peptoniphilus lacrimalis*	Firmicutes	Peptoniphilaceae	*Peptoniphilus*
*Peptostreptococcus anaerobius*	Firmicutes	Peptoniphilaceae	*Peptostreptococcus*
*Porphyromonas asaccharolytica*	Bacteroidetes	Porphyromonadaceae	*Porphyromonas*
*Porphyromonas bennonis*	Bacteroidetes	Porphyromonadaceae	*Porphyromonas*
*Porphyromonas somerae*	Bacteroidetes	Porphyromonadaceae	*Porphyromonas*
*Prevotella bergensis*	Bacteroidetes	Prevotellaceae	*Prevotella*
*Prevotella bivia*	Bacteroidetes	Prevotellaceae	*Prevotella*
*Prevotella buccalis*	Bacteroidetes	Prevotellaceae	*Prevotella*
*Prevotella colorans*	Bacteroidetes	Prevotellaceae	*Prevotella*
*Prevotella corporis*	Bacteroidetes	Prevotellaceae	*Prevotella*
*Prevotella disiens*	Bacteroidetes	Prevotellaceae	*Prevotella*
*Prevotella timonensis*	Bacteroidetes	Prevotellaceae	*Prevotella*
*Propionimicrobium lymphophilum*	Actinobacteria	Propionibacteriaceae	*Propionimicrobium*
*Ralstonia insidiosa*	Proteobacteria	Burkholderiaceae	*Ralstonia*
*Slackia exigua*	Actinobacteria	Eggerthellaceae	*Slackia*
*Staphylococcus capitis*	Firmicutes	Staphylococcaceae	*Staphylococcus*
*Staphylococcus cohnii*	Firmicutes	Staphylococcaceae	*Staphylococcus*
*Staphylococcus epidermidis*	Firmicutes	Staphylococcaceae	*Staphylococcus*
*Staphylococcus haemolyticus*	Firmicutes	Staphylococcaceae	*Staphylococcus*
*Staphylococcus hominis*	Firmicutes	Staphylococcaceae	*Staphylococcus*
*Staphylococcus simulans*	Firmicutes	Staphylococcaceae	*Staphylococcus*
*Staphylococcus warneri*	Firmicutes	Staphylococcaceae	*Staphylococcus*
*Streptococcus agalactiae*	Firmicutes	Streptococcaceae	*Streptococcus*
*Streptococcus anginosus*	Firmicutes	Streptococcaceae	*Streptococcus*
*Streptococcus parasanguinis*	Firmicutes	Streptococcaceae	*Streptococcus*
*Streptococcus salivarius*	Firmicutes	Streptococcaceae	*Streptococcus*
*Trueperella bernardiae*	Actinobacteria	Actinomycetaceae	*Trueperella*
*Ureaplasma sp.*	Tenericutes	Mycoplasmataceae	*Ureaplasma*
*Varibaculum cambriense*	Actinobacteria	Actinomycetaceae	*Varibaculum*
*Veillonella atypica*	Firmicutes	Veillonellaceae	*Veillonella*
*Veillonella paravula*	Firmicutes	Veillonellaceae	*Veillonella*
*Veillonella ratti*	Firmicutes	Veillonellaceae	*Veillonella*

**Table 3 ijms-23-12212-t003:** Parameter estimates and confidence intervals found per incubation technique.

Incubation Method	Mean Number of Species Found $\left(\hat{\lambda }\right)$	95% Confidence Interval		Mean Number of Unique * Species Found $\left(\hat{\lambda }\right)$	95% Confidence Interval		Mean Number of Obligate Anaerobes Found $\left(\hat{\lambda }\right)$	95% Confidence Interval	
		Lower Bound	Upper Bound		Lower Bound	Upper Bound		Lower Bound	Upper Bound
*Aerobic*	7.499	4.310	12.633	3.435	1.976	5.640	1.085	0.146	4.249
*Anaerobic*	6.788	3.889	11.474	3.268	1.872	5.386	0.700	0.093	2.800
*No pre-incubation*	7.025	4.029	11.861	3.771	2.183	6.147	1.304	0.176	5.092

* The number of species in the sample not found by the other two incubation-methods within the same sample. Aerobic stands for the aerobic pre-incubation, anaerobic stands for the anaerobic pre-incubation.

**Table 4 ijms-23-12212-t004:** Summary of all of the used culture media, incubation circumstances, composition and targeted species.

Full Name (Abbreviation)	Incubation	Composition/L or Brand	Targeted Species
Blood agar (HEM)	Aerobic + 5% CO_2_, 37 °C	40 g tryptic soy agar Oxoid 2 mL 1% haemin solution 2 mL nicotinamide dinucleotide 50 mL horse blood	Nonselective medium for aerobic bacteria
Chocolate agar PolyViteX (VCAT)	Aerobic + 5% CO_2_, 37 °C	Biomerieux (*REF. 43611*)	Selective medium for *Neisseria gonorrhoeae* and *Neisseria meningitidis.*
MacConkey agar (MAC)	Aerobic, 37 °C	Biomerieux (*REF. 43141*)	Selective medium for gram-negative aerobic bacteria (due to presence of crystal violet)
Sabouraud agar with gentamicin and chloramphenicol (SAB)	Aerobic, 37 °C	BD (*REF. 254041*)	Nonselective medium for fungi and yeast
Anaerobic agar (ANA)	Anaerobic, 37 °C	- 46 g anaerobe agar Neogen - 50 mL horse blood	Nonselective medium for anaerobic bacteria
Selective anaerobic agar (SANA)	Anaerobic, 37 °C	46 g anaerobe agar Neogen 0.01 g nalidixic acid 1.05 mL tween 80 50 mL hemolysed horse blood	Selective medium for gram-positive anaerobic bacteria
Schaedler anaerobic agar with sheep blood, vitamine K1 and haemin (SCHAED)	Anaerobic, 37 °C	Thermo Scientific (*REF. 10453833*)	Nonselective and highly nutritious medium for fastidious anaerobic bacteria
Ureaplasma/mycoplasma agar (UUA)	Anaerobic, 37 °C	24.75 g tryptic soy broth (Neogen), 12 g anaerobe agar (Neogen), 0.16 g manganese(II) sulphate (Merck), 200.0 mL horse serum (TICO), 5.0 mL penicillin G solution, 5.0 mL VITOX enrichment (Oxoid), 10.0 mL 10% urea solution, 2.5 mL 4% L-cysteine solution, 17.5 mL baker’s yeast extract (25% fresh), 825.0 mL aqua purificata	Selective medium for *Mycoplasma* and *Ureaplasma* sp.
Ureaplasma broth	Aerobic, 37 °C	8.4 g PPLO without crystal violet (BD), 0.4 g yeast extract (BD), 4.0 mL bromothymol blue, 40.0 mL horse serum (Biotech), 1.0 mL 1% urea solution, 2.0 mL Clamoxyl (400 mg/2 mL), 2.0 mL Nystatin (20,000 IE/2 mL), 360.0 mL aqua purificata	Selective broth for *Ureaplasma* sp.
Mycoplasma broth	Aerobic, 37 °C	1.4 g PPLO without crystal violet (BD), 4.0 g tryptone (BD), 3.2 g peptone (Biokar), 2.0 g L-arginine (Acros), 200.0 mL CMRL medium (Gibco), 40.0 mL 2% yeastolate (Gibco), 68.0 mL horse serum (Biotech), 4.0 mL glutamine, 14.0 mL baker’s yeast extract solution, 4.0 mL 0.2% phenol red, 6.13 mL 7.5% sodium bicarbonate, 2.0 mL Clamoxyl (400 mg/2 mL), 2.0 mL Nystatin (20,000 IE/2 mL), 52.0 mL aqua purificata	Selective broth for *Mycoplasma* sp.

**Table 5 ijms-23-12212-t005:** Summary of the WASP^®^ settings and used incubators. For this protocol, we used the five-fold dilution-striking pattern with one µL volume. The CO_2_ incubator and non-CO_2_ aerobic incubator from our WASPLab^®^ system were automatically used after the inoculation. * Due the fact that WASPLab^®^ do not offer (yet) an anaerobic incubator, these plates had to be placed manually in our anaerobic incubator after inoculating with the WASP^®^.

Settings WASP^®^ Module
Inoculation pattern	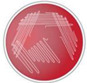
Inoculation volume	1 µL
**Settings adjoined incubator**
CO_2_ incubator	Storage of HEM and VCAT plates
Non-CO_2_ incubator	Storage of SAB and MAC plates
*Anaerobe incubator**	*Storage of ANA, SANA and Schaedler plates*

**Table 6 ijms-23-12212-t006:** Composition of the home-made supplement mix we added to all of the the pre-incubation bottles. The composition is based on the YCFA and R-medium previously used by Didier Raoult et al. [14]. The first 12 ingredients are sterilized through an autoclave. The other four ingredients are thermolabile and were sterilized through filtration. Sterility was checked at the end of the process.

Tryptone	10 g
Proteose peptone	10 g
Yeast extract	10 g
Glucose	10 g
Natrium chloride	5 g
L-cysteine	1 g
Potassium hydrogen phosphate	0.45 g
Potassium dihydrogen phosphate	0.45 g
Haemin	0.1 g
Calcium chloride dihydrate	0.09 g
Magnesium sulphate heptahydrate	0.045 g
Distilled water	Ad 1 L
Thiamine	0.01 g
Riboflavin	0.01 g
Biotin	0.004 g
Folic acid	0.004 g
Distilled water	Ad 5 mL

## Data Availability

The datasets generated and/or analyzed during the current study are available in the National Library of Medicine repository. GenBank submission: SUB11956655 prokaryotic 16S rRNA/16S rRNA endometrium.

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
