# Peer review of "Culturomics to Investigate the Endometrial Microbiome: Proof-of-Concept"

_ijms, 2022, doi:10.3390/ijms232012212_

Round 1

Reviewer 1 Report

Infertility represents a great provocation in our age. This study greatly contributes to the foundation laying effort for identifying the endometrial microbiota involved in fertility. The great amount of labor involved and the fine results validating the methods recommend this article for publication. Also, it produces material for questioning and further evolvement in the matter.

Author Response

Infertility represents a great provocation in our age. This study greatly contributes to the foundation laying effort for identifying the endometrial microbiota involved in fertility. The great amount of labor involved and the fine results validating the methods recommend this article for publication. Also, it produces material for questioning and further evolvement in the matter.

We thank the reviewer for taking the time and effort to review our manuscript.

Reviewer 2 Report

Authors propose and describe in details the improved culture based methodology for assessment of endometrial microbiota. This approach is rightly stated and shown to potentially overcome biases of prior culturomics studies applied to microbioma in general, in particular that of reproductive tract, consisting in metagenomics and 16S amplicon analysis.

Having stated that, i have several remarks that would require attention of the authors  prior to the approval of this manuscript for publishing 

I think that a title is somewhat misleading - leading a reader to think that the culturomics is a method that enabled to link the endometrial microbioma to a specific clinical condition - a subfertility, while this is not a case. This of course is one of the important applications of the method, but the article itself is not designed, or not even meant , to provide such an evidence. SO there is no link to subfertility here, if not for a selection of samples for a proof of concept, Therefore I would just state culturomics as a method to investigate endometrial microbioma in general - in health and in disease, not linked to subfertility YET.

Indeed the first (obvious) observation is the lack of "normal" fertile subjects derived samples in this study. Only in such a case there would be rational to talk about "describing the microbioma profile in subfertile women"!  Therefore if for any (etical or other) reasons the authors did not include such comparison in the study, they should avoid some statements specifically linked to subfertility or they should reference some independent data that show or hint that the profile hereby described is different of characteristic of subfertility! 

I might have missed this but - what are the negative controls in your approach? please highlight this

In Figure 3 and 4 the literature derived data on microbioma used for comparison with your data comes from metagenomics approach onto endometrial biopsies? on which kind of subjects? infertile? please clarify this also in the legend and in the result!

One question on the rational for design and selection of culture conditions used to identify differently rapresented and differently rapid/slow groing strains. Do the conditions used to culture inoculates resamble endometrium environment and its changes along the physiological and pathological conditions?  as reported for instance in https://academic.oup.com/humupd/article/24/1/15/4565549 oxygen, temperature and pH dynamics in endometrium reflect cyclic changes dependent on the cyclic phase, age, hormons, muscle integrity and so on. While some of these conditions such as temperature and PH can be reproduced and challenged in vitro/ex vivo, these issue merit some extensive considerations by the authors in discussion. Of course it would be good to understand if other than medium composition and time of incubation factors were or can be included in the current of future study design.

what about menstrual cycle phase of donors? Are they all syncronized and homogeneous? 

for discussion please consider also the following ( I do not have any interest whatsoever to quote this papers)

https://www.sciencedirect.com/science/article/pii/S0165037822001632

https://www.frontiersin.org/articles/10.3389/fimmu.2019.02387/full

https://rep.bioscientifica.com/view/journals/rep/163/5/REP-21-0120.xml

what about menstrual cycle phase of donors? Are they all syncronized and homogeneous? 

The authors rightly state that the potential drawback of this study is a control of contamination with low tract (vaginal) microbioma. Two quistions: how continuous - transferable this microbioma is along the tract normally or it is confined by different environments in different tract compartments? Second, wouldn't have been good and correct to test also the vaginal microbioma from the same donors with the same technique? Can you please comment on this? 

Second neglected issue even at the level of discussion  is the blood microbiome. Can this infiltration into endometrium change cyclically or with some conditions such as inflammation or conception and implantation? a part from suggestion to address blood microbioma empirically, this issue deserves the atttention in this manuscript.

it is not clear if culturomics over metagenomics based approaches ensure additional quantitative information for example on the prevalence and ratios of different strains within microbiota. Can you comment on this

Therefore I think that this paper clearly presents the potential and approach to expand our understanding of the microbioma composition in some difficult samples such as endometrium ones featuring either complementary approach with respect to metagenomics for example, or alternative that overcomes some of very well explained limitations and biases. On the other hand it has important implications for futher studies but some statements can not be made in this manuscript due to a current design.

what is missing is also a step versus realistic mapping of endometrium microbioma via culturomix. is there some more feasible and non invasive way to get get the endometrium sample or a surrogate of this one? 

I do not know whether it is feasible to authors to add some experimental points (samples or conditions) to the current study, but the impact of  the manuscript would be much improved by more accurate, comprehensive and acute discussion and inclusion of prior literature and concepts

Author Response

Reviewer Comments

Reviewer 2

Authors propose and describe in details the improved culture based methodology for assessment of endometrial microbiota. This approach is rightly stated and shown to potentially overcome biases of prior culturomics studies applied to microbioma in general, in particular that of reproductive tract, consisting in metagenomics and 16S amplicon analysis. Having stated that, i have several remarks that would require attention of the authors  prior to the approval of this manuscript for publishing 

We thank the reviewer for the suggestions and improvements. Through the implementations and adaptations, we believe that the manuscript has improved significantly.

1) I think that a title is somewhat misleading - leading a reader to think that the culturomics is a method that enabled to link the endometrial microbioma to a specific clinical condition - a subfertility, while this is not a case. This of course is one of the important applications of the method, but the article itself is not designed, or not even meant , to provide such an evidence. SO there is no link to subfertility here, if not for a selection of samples for a proof of concept, Therefore I would just state culturomics as a method to investigate endometrial microbioma in general - in health and in disease, not linked to subfertility YET.

We agree with this comment. Therefore, we changed our title to ‘Culturomics to investigate the endometrial microbiome: proof-of-concept’

2) Indeed the first (obvious) observation is the lack of "normal" fertile subjects derived samples in this study. Only in such a case there would be rational to talk about "describing the microbioma profile in subfertile women"!  Therefore if for any (etical or other) reasons the authors did not include such comparison in the study, they should avoid some statements specifically linked to subfertility or they should reference some independent data that show or hint that the profile hereby described is different of characteristic of subfertility! 

It is a fact that this study was performed in subfertile women, undergoing diagnostic hysteroscopy followed by endometrial biopsy as part of routine work-up. For ethical reasons, we could not include “normal fertile” women at this stage of our research. However, this does not lead to a statement that our results are specifically linked to subfertility

We agree with the reviewer’s suggestion, therefore we modified the manuscript and deleted ‘subfertile’ from the title and from line 115. We deleted ‘healthy’ from line 318. As such, the manuscript describes the culturomics approach in a more objective, non-clinically oriented manner.

3) I might have missed this but - what are the negative controls in your approach? please highlight this

Thank you for pointing this out. We highlighted the negative controls in lines 173-176.

Blank aerobic and anaerobic supplemented blood culture bottles, both supplemented with 2.0 mL sterile rumen fluid, 2.0 mL sterile defibrinated sheep blood and 2.0 mL of our homemade supplement mix, accompanied every sample throughout the whole process to ensure sterility of the used substances.

4) In Figure 3 and 4 the literature derived data on microbioma used for comparison with your data comes from metagenomics approach onto endometrial biopsies? on which kind of subjects? infertile? please clarify this also in the legend and in the result!

The data we used from the literature contains a very heterogeneous group of subjects. Indeed, this group consists mainly of subfertile and infertile patients. A detailed description of the subjects can be found in sources 7 and 23.

We added ‘in subfertile and infertile women’ in lines 272-273. We added ‘Literature overview includes both infertile and subfertile women.’ in the legend of figure 3 and 4.

5) One question on the rational for design and selection of culture conditions used to identify differently rapresented and differently rapid/slow groing strains. Do the conditions used to culture inoculates resamble endometrium environment and its changes along the physiological and pathological conditions?  as reported for instance in https://academic.oup.com/humupd/article/24/1/15/4565549 oxygen, temperature and pH dynamics in endometrium reflect cyclic changes dependent on the cyclic phase, age, hormons, muscle integrity and so on. While some of these conditions such as temperature and PH can be reproduced and challenged in vitro/ex vivo, these issue merit some extensive considerations by the authors in discussion. Of course it would be good to understand if other than medium composition and time of incubation factors were or can be included in the current of future study design. For discussion please consider also the following ( I do not have any interest whatsoever to quote this papers)

The used culture conditions were selected based on previous studies performed on stool samples. We selected the richest media in this study in order to give as many species as possible a chance to grow. However, no attention was paid to the physico-chemical properties of the endometrium. We think this would indeed be interesting to take this into account in subsequent studies. We added this statement with reference https://academic.oup.com/humupd/article/24/1/15/4565549 in lines 326-332.

https://www.sciencedirect.com/science/article/pii/S0165037822001632

Interesting paper. Indeed, it would be interesting to highlight the plasticity of the uterine microbiota during the menstrual cycles. We therefore added a statement with this reference in lines 332-334.

https://www.frontiersin.org/articles/10.3389/fimmu.2019.02387/full

https://rep.bioscientifica.com/view/journals/rep/163/5/REP-21-0120.xml

We think the other two suggested papers are very interesting. They do not, in our view, fit entirely within the scope of this paper’s discussion. Since we are approaching this manuscript rather technically, we would like not to dwell too far on clinical parameters like diseases and immunity.

6) what about menstrual cycle phase of donors? Are they all syncronized and homogeneous? 

Yes, all biopsies were taken during the follicular phase. We highlighted this in lines 129-130.

7) The authors rightly state that the potential drawback of this study is a control of contamination with low tract (vaginal) microbioma. Two quistions: how continuous - transferable this microbioma is along the tract normally or it is confined by different environments in different tract compartments? Second, wouldn't have been good and correct to test also the vaginal microbioma from the same donors with the same technique? Can you please comment on this? 

Accumulating evidence demonstrates that the uterogenital tract is an open system with a microbiota continuum that gradually changes from the outer to the inner organs, with decreasing bacterial abundance and increasing bacterial diversity from the vagina to the ovaries. Since the pH in the vagina is much lower than the pH in the endometrium, this seems logical. We added lines 52-55 and reference “Endometrial microbiota composition is associated with reproductive outcome in infertile patients. Microbiome 10, 1 (2022). https://doi.org/10.1186/s40168-021-01184-w. Moreno, I., Garcia-Grau, I., Perez-Villaroya, D. et al. »

Indeed, it would have been a good idea to include vaginal samples from the same donors. Since the composition of microbiota are dynamic, it would not be correct to retrospectively include vaginal samples from these 10 patients. We, therefore, will investigate this relationship in a follow-up study.

8) Second neglected issue even at the level of discussion  is the blood microbiome. Can this infiltration into endometrium change cyclically or with some conditions such as inflammation or conception and implantation? a part from suggestion to address blood microbioma empirically, this issue deserves the atttention in this manuscript.

This is an interesting suggestion of the reviewer. To the best of our knowledge, there are no studies providing evidence on the relationship between these microbiota. However, we believe that the blood microbiome might influence the endometrial microbiome. For example, by transporting species from another mucosa (oral, gut,…) to the endometrium. We addressed this in the discussion and added lines 348-350 and added reference “The Healthy Human Blood Microbiome: Fact or Fiction? Front. Cell. Infect? Microbio. 08 May 2019, https://doi.org/10.3389/fcimb.2019.00148. Diego J. Castillo, Riaan F. Rifkin, Don A. Cowan and Marnie Potgieter.”

9) it is not clear if culturomics over metagenomics based approaches ensure additional quantitative information for example on the prevalence and ratios of different strains within microbiota. Can you comment on this

Thank you for pointing this out. Culturomics provides only qualitative information, where sequencing approaches also give a representation of the quantitative composition of microbiota. We addressed this in lines 97-99.

10) what is missing is also a step versus realistic mapping of endometrium microbioma via culturomix. is there some more feasible and non invasive way to get get the endometrium sample or a surrogate of this one? 

The pipelle biopsy (inner-outer catheter) is considered the least invasive way to obtain endometrial tissue. This is often used in routine fertility procedures and is well tolerated by the patient. The uteral cavity could also be sampled with an empty embryo transfer catheter, but then one would not obtain real tissue material and the biomass becomes even more limiting.

11) I do not know whether it is feasible to authors to add some experimental points (samples or conditions) to the current study, but the impact of  the manuscript would be much improved by more accurate, comprehensive and acute discussion and inclusion of prior literature and concepts

Due to the huge workload, we deliberately chose to limit both the number of samples and conditions in this proof-of-concept. As described in lines 334-336, the inclusion of additional conditions is not directly proportional to the discovery of additional species. However, conditions that take the physico-chemical properties into account, within the endometrium could be of added value in future studies. We highlighted this in lines 329-334.

Therefore I think that this paper clearly presents the potential and approach to expand our understanding of the microbioma composition in some difficult samples such as endometrium ones featuring either complementary approach with respect to metagenomics for example, or alternative that overcomes some of very well explained limitations and biases. On the other hand it has important implications for futher studies but some statements can not be made in this manuscript due to a current design.

We agree with the reviewer and have adapted the manuscript accordingly. We have removed and rephrased all passages, referring to fertility issues. Yet, as mentioned in our answer to comment 2, even without making statements, we cannot avoid that a lector should draw conclusions from the fact we selected only subfertile women here. It is however not our intention.

Round 2

Reviewer 2 Report

I beleive that the revisions introduced by the authoresimproved the discussion and paved the way to interesting follow on studies. I still miss some controls and experiments, might be my  subjective idea on a study design, but can also understand why including them in the present study can be challenginy.  I think overall that this study is very interesting and it leads to the proceedings in several fields.